# New Insights into the Neuromyogenic Spectrum of a Gain of Function Mutation in SPTLC1

**DOI:** 10.3390/genes13050893

**Published:** 2022-05-17

**Authors:** Heike Kölbel, Florian Kraft, Andreas Hentschel, Artur Czech, Andrea Gangfuss, Payam Mohassel, Chi Nguyen, Werner Stenzel, Ulrike Schara-Schmidt, Corinna Preuße, Andreas Roos

**Affiliations:** 1Department of Paediatric Neurology, Center for Neuromuscular Disorders in Children and Adolescents, Center for Translational Neuro- and Behavioral Sciences, University Clinic Essen, University of Duisburg-Essen, 45122 Essen, Germany; andrea.gangfuss@uk-essen.de (A.G.); ulrike.schara-schmidt@uk-essen.de (U.S.-S.); andreas.roos@uk-essen.de (A.R.); 2Institute of Human Genetics, Medical Faculty, RWTH Aachen University, 52062 Aachen, Germany; fkraft@ukaachen.de; 3Leibniz-Institut für Analytische Wissenschaften—ISAS—e.V., 44139 Dortmund, Germany; andreas.hentschel@isas.de (A.H.); artur.czech@gmx.net (A.C.); chi.nguyen@isas.de (C.N.); 4Neuromuscular and Neurogenetic Disorders of Childhood Section, National Institute of Neurological Disorders and Stroke, National Institutes of Health, 10 Center Dr., Bethesda, MD 20892, USA; payam.mohassel@nih.gov; 5Department of Neuropathology, Charité-Universitätsmedizin Berlin, Corporate Member of Freie Universität Berlin and Humboldt-Universität zu Berlin, 10117 Berlin, Germany; werner.stenzel@charite.de (W.S.); corinna.preusse@charite.de (C.P.); 6Department of Neurology with Institute for Translational Neurology, University Hospital Münster, 48149 Münster, Germany; 7Children’s Hospital of Eastern Ontario Research Institute, University of Ottawa, Ottawa, ON K1H 8L1, Canada; 8Department of Neurology, Heimer Institute for Muscle Research, University Hospital Bergmannsheil, Ruhr-University Bochum, 44789 Bochum, Germany

**Keywords:** *SPTLC1*, juvenile ALS, HSAN1A, proteomic profiling, serine palmitoyltransferase

## Abstract

*Serine palmitoyltransferase long chain base subunit 1* (*SPTLC1*) encodes a serine palmitoyltransferase (SPT) resident in the endoplasmic reticulum (ER). Pathological *SPTLC1* variants cause a form of hereditary sensory and autonomic neuropathy (HSAN1A), and have recently been linked to unrestrained sphingoid base synthesis, causing a monogenic form of amyotrophic lateral sclerosis (ALS). It was postulated that the phenotypes associated with dominant variants in *SPTLC1* may represent a continuum between neuropathy and ALS in some cases, complicated by additional symptoms such as cognitive impairment. A biochemical explanation for this clinical observation does not exist. By performing proteomic profiling on immortalized lymphoblastoid cells derived from one patient harbouring an alanine to serine amino acid substitution at position 20, we identified a subset of dysregulated proteins playing significant roles in neuronal homeostasis and might have a potential impact on the manifestation of symptoms. Notably, the identified p.(A20S)-SPTLC1 variant is associated with decrease of transcript and protein level. Moreover, we describe associated muscle pathology findings, including signs of mild inflammation accompanied by dysregulation of respective markers on both the protein and transcript levels. By performing coherent anti-Stokes Raman scattering microscopy, presence of protein and lipid aggregates could be excluded.

## 1. Introduction

SPTLC1 encodes for a member of the serine palmitoyltransferase (SPT) family, forming a heterodimer with SPTLC2 or SPTLC3, and thus constitutes the catalytic core of the protein complex [1]. The composition of the serine palmitoyltransferase (SPT) complex determines the substrate preference: the SPTLC1-SPTLC2-SPT small subunit A (SPTSSA) complex indicates a strong preference for C16-CoA substrate, while the SPTLC1-SPTLC3-SPTSSA isozyme uses both, C14-CoA and C16-CoA are substrates with a slight preference for C14-Coenzyme (CoA) [1]. The complex is localized to the endoplasmic reticulum (ER) [2] and expressed at various abundances across the neuromuscular axis and within cultured fibroblasts and blood cells (Appendix A).

Notably, dominant variants are causative for an axonal form of hereditary sensory and autonomic neuropathy (HSAN1A: MIM#162400) with onset in the second or third decades of life. Initial symptoms are loss of pain, touch, heat, and cold sensation over the feet, followed by distal muscle wasting and weakness [3,4]. Loss of pain sensation may lead to chronic skin ulcers and distal amputations [3,4]. Smaller studies indicated the efficacy of L-serine for SPTLC1-related HSAN [5]. Whereas manifestation of the HSAN1A phenotype was linked to a loss of SPTLC1 function based on the pathogenic variants, recently dominant variants were linked to childhood onset of amyotrophic lateral sclerosis based on a pathological gain of SPTLC1 function. These variants disrupt the normal homeostatic SPT regulation, resulting in unregulated activity and elevated levels of canonical SPT products. Pathophysiologically, this is in contrast with pathogenicity of HSAN1A-associated SPTLC1 variants. In our patient, a shift of SPT amino acid usage from serine to alanine resulting in elevated levels of deoxysphingolipids was already described, in addition to a detailed description of the pathogenicity [6]. However, the vulnerability of motor neurons is in line with the ubiquitous expression of the protein including the proven expression of SPTLC1 in motor neurons of healthy spinal cord [7].

It was postulated that phenotypes associated with dominant variants in SPTLC1 may represent a continuum between sensory neuropathy and ALS, complicated by additional symptoms such as cognitive impairment and retina pathology. Future (post-mortem) studies that determine the central nervous system pathology (e.g., TDP43 and tau protein deposition) underlying the motor neuron deficits and the cognitive impairment may resolve the nature of the overlap with other neurodegenerative diseases [7]. By performing unbiased proteomic profiling on immortalized lymphoblastoid cells derived from an SPTLC1-patient with juvenile-onset of ALS without brain abnormalities and normal cognitive function, we identified a set of proteins which might contribute to manifestation of certain symptoms of SPTLC1-pathology upon an individual expression level. The results of our proteomic studies demonstrate that the alanine to serine amino acid substitution at position 20 leads to a 50% decrease of SPTLC1 protein level. Microscopic investigations on the quadriceps muscle derived from our patient revealed a neurogenic atrophy characterized by fiber-type grouping and accompanied by unspecific subsarcolemmal enrichment of mitochondria and unravelled presence of mild inflammation. Latter findings were confirmed by the results of comprehensive quantitative transcript studies.

## 2. Materials and Methods

### 2.1. Generation of Immortalized Lymphoblastoid Cells

Peripheral blood was drawn in Na^+^/Heparin tubes and transferred to a 50 mL conical centrifuge tube. Lysis of erythrocytes was achieved by adding lysis buffer (150 mM ammonium chloride, 10 mM potassium hydrogencarbonat, and 0.1 mM EDTA) with 30 min incubation at 4 °C. Cells were spun down and the supernatant was discarded. The lysis step was repeated, and the cells were resuspended in phosphate-buffered saline (PBS). After another centrifugation step, the cell pellet was resuspended in 3 mL virus-containing supernatant of B95-8 cells (RPMI media with 20% FCS and 200 ng/mL cyclosporin). 24 h later, 7 mL media (RPMI with 20% FCS and 200 ng/mL cyclosporin) were added. Cells were grown for several weeks until transformation was achieved. During the transformation period, 5 mL of media were taken and 5 mL of fresh media were added every week without destroying the cell clumps. Transformed lymphocytes were maintained at 0.5–2 million cells per ml in RPMI media supplemented with 10% FCS.

### 2.2. Proteomic Profiling of Immortalized Lymphoblastoid Cells

#### 2.2.1. Sample Preparation for Mass Spectrometry

The snap-frozen samples were lysed by adding 200 µL of 50 mM Tris-HCl (pH 7.8) buffer, 5% SDS, and cOmplete ULTRA protease inhibitor (Roche, Basel, Switzerland) using the Bioruptor^®^ (Diagenode, Denville, NJ, USA) for 10 min (30 s on, 30 s off, 10 cycles) at 4 °C, followed by centrifugation at 4 °C and 20,000× *g* for 15 min. A BCA assay was used to determine the protein concentration in the supernatant according to the manufacturer’s instructions. Free sulfhydryl bonds were alkylated with 15 mM IAA at room temperature (RT) in the dark for 30 min, after disulphide bonds were reduced with 10 mM TCEP at 37 °C for 30 min. Proteolysis was performed on 100 µg of protein from each sample using the S-Trap procedure (Protifi, Farmingdale, NY, USA) with a protein to trypsin ratio of 20:1. The trypsin incubation period was increased to 2 h at 37 °C. FA was used to stop proteolysis (pH < 3.0).

After desalting, all proteolytic digests were checked for complete digestion using monolithic column separation (PepSwift monolithic PS-DVB PL-CAP200-PM, Dionex, Germering, Germany) on an inert Ultimate 3000 HPLC (Dionex) by direct injection of a sample of 1 µg. At a flow rate of 2.2 µL/min and 60 °C, a binary gradient (solvent A: 0.1% TFA, solvent B: 0.08% TFA, 84% ACN) ranging from 5–12% B in 5 min and then from 12–50% B in 15 min was used. UV traces were taken at a wavelength of 214 nm [8].

#### 2.2.2. Proteomic Analysis

Samples were analyzed using an UltiMate 3000 RSLC nano UHPLC connected to a QExactive HF mass spectrometer with the total amount of peptide administered always being 1 µg. The samples were first separated on a 75 m × 50 cm, 100, C18 main column with a flow rate of 250 nL/min and a linear gradient consisting of solution A (99.9% water, 0.1% formic acid) and solution B (84% acetonitrile, 15.9% water, 0.1% formic acid) with a pure gradient length of 120 min (3–45% solution B). The gradient was composed as follows: 3% B for 20 min, 3–35% for 120 min, followed by three wash steps, each lasting 3 min and ranging from 95% buffer B to 3% buffer A. The instrument was left to equilibrate for 20 min after the final washing procedure. MS data were acquired utilizing an in-house generated spectrum library in a data independent acquisition (DIA) mode. Each sample was combined with an adequate amount of iRT standard before examination (Biognosys, Schlieren, Switzerland). Full MS scans were obtained at a resolution of 60,000 (Orbitrap) from 300–1100 *m*/*z*, with the polysiloxane ion at 445.12002 *m*/*z* serving as the lock mass. Maximum injection time was set to 20 milliseconds and the automatic gain control (AGC) was set to 3E6. Following the full MS scans, 23 DIA windows were acquired at a resolution of 30,000 (Orbitrap) with an AGC of 3E6 and an nCE of 27. Each DIA window covered a range of 28 *m*/*z* with 1 *m*/*z* overlap, starting at 400 *m*/*z*. For proteomic analysis, samples from two healthy individuals and the sample from the patient were processed. After the preparation of immortalized lymphoblastoid (see Section 2.1), samples were propagated (cell culture) to 2 × 3 samples for controls and 1 × 3 samples for the patient. In total, we processed 9 samples independently in our proteomics sample preparation and analysis pipeline.

#### 2.2.3. Data Analysis

Data were imported into Spectronaut software (Biognosys) and analysed with a library-based search, making use of a spectral library produced in-house. Default search and extraction settings were used (BGS Factory settings). The human proteome data was obtained from UniProt (www.uniprot.org, accessed on 23 July 2018)), which contains a total of 20,374 entries.

Only proteins identified with at least two unique peptides were selected for further analysis to achieve reliable label-free quantification. Average normalized abundances (obtained using Spectronaut) were calculated for each protein. Ratios between patient samples and corresponding controls were calculated. Finally, MS Excel was used to calculate student’s *t*-test *p*-values for each protein. Based on the exported data, a log 10 transformation of the LFQ data and a log 2 transformation of the calculated ratios were performed using MS Excel to visualize the results.

#### 2.2.4. In Silico Studies of Proteomic Findings

To obtain a detailed overview of the biological processes, a Gene Ontology (GO)-term assessment was performed. This analysis was carried out using the web software DAVID (Database for Annotation, Visualization, and Integrated Discovery) [9,10]. Only proteins with positive or negative regulation that were significantly regulated (*p*-value of 0.05 or less) were included in the analysis. Results were manually selected for relevant outcomes using the GO-Terms for biological process, molecular function, and cellular components.

### 2.3. Morphological Analysis of Quadriceps Muscle

All stains were performed on 8 µm cryostat sections, according to standard procedures. Immunohistochemical reactions were carried out as described previously [11]. The following antibodies were used for staining procedures: C5b-9 (DAKO, clone aE11, 1:200), CD3 (Novocastra, clone PS1, 1:50), CD20 (DAKO, clone L26, 1:200), CD68 (DAKO, clone EBM1, 1:100), CD138 (DAKO, clone MI15, 1:30), MHC-cl. I (DAKO, clone W6/32, 1:1000), MHC-cl. II (DAKO, clone CR3/43, 1:100), MyHCdev (Novocastra, clone RNMY2/9D2, 1:5), MyHCneo (Novocastra, clone NB-MHCn, 1:20) MHC-Slow (Novocastra, clone WB-MHCS, 1:100), and MHC-Fast (Novocastra, clone WB-MHCF, 1:100). For the studies on muscle, two age- and sex-matched controls were used.

### 2.4. Transcript Studies on Quadriceps Muscle

Total RNA was extracted from muscle specimens using the technique described previously [12]. Briefly, cDNA was synthesized using the High-Capacity cDNA Archive Kit (Applied Biosystems, Foster City, CA, USA). For qPCR reactions, 10 ng of cDNA were used for analysis using the QuantStudio 6 Flex System (Applied Biosystems, Foster City, CA, USA). Running conditions were: 95 °C for 0:20, 95 °C for 0:01 and 60 °C for 0:20, 45 cycles (values above 40 cycles were defined as not expressed). All targeted transcripts were investigated as triplicates including PGK1 as a reference. The qPCR assay identification numbers, TaqMan^®^ Gene Exp Assay from Life Technologies (Carlsbad, CA, USA)/ThermoFisher (Waltham, MA, USA) are listed as the following: HLA-DOB Hs00157950_m1, HLA-DPB Hs03045105_m1, HLA-DRA Hs00219575_m1, HLA-DRB Hs02339733_m1, STAT1 Hs01013989_m1, STAT2 Hs01013123_m1, IL1B Hs01555410_m1, CD206 (MRC1) Hs00267207_m1, and PGK1 Hs99999906_m1. The fold change (2^−ΔΔCT^) is displayed for the patient’s gene expression compared to non-disease controls (NDCs).

### 2.5. Coherent Anti-Stokes Raman Scattering (CARS) and Statistical Evaluation of Muscle Fiber Calibres

5 µm cryosections were generated for the coherent anti-Stokes Raman scattering (CARS) microscopy and stored at −80 °C. For CARS and second harmonic generation (SHG) measurements, samples were dried under a constant nitrogen gas flow at room temperature and no further sample preparation was applied.

All spectroscopic measurements were performed on a modified Leica TCS SP 8 CARS laser scanning microscope [13]. Generated CARS signals were detected in forward direction. A 40× water immersion objective (IRAPO 40×/1.10 WATER) was used for CARS imaging. CARS spectra were acquired for a field of 291 × 291 µm (2048 × 2048 pixel) by tuning the pump laser from 804.0 nm to 826.4 nm with a step size of 0.7 nm, a pixel dwell time of approximately 10 µs, and averaging of two images.

Leica software LAS X (Version 2.0) was used for manual determination of the muscle fibers from the FCARS images. For cross-sectioned muscle samples, the length and width of fully imaged fibers was determined, and the caliber averaged from the two values. A total of 155 fibers were analyzed for the SPTLC1 patient. As reference, 1457 fibers were analyzed from five control samples.

### 2.6. SPTLC1 mRNA Expression and Transcript Studies

RNA was isolated from EBV-immortalized lymphocytes with peqGOLD TriFast™ (30–2010, VWR International, Radnor, PA, USA) according to the manufacturer’s instructions. 2 µg RNA was used for cDNA synthesis with Maxima™ First Strand cDNA Synthesis Kit (K1641, Thermo Fisher Scientific, Waltham, MA, USA) following the manufacturer’s protocol. 25 ng cDNA was applied for qPCR with PowerUp™ SYBR^®^ Green according to manufacturer’s protocol using the following cycling protocol: 95 °C for 10:00; 95 °C for 0:05 and 60 °C for 0:30, 40 cycles. Specificity of the PCR products were confirmed by melting curve analysis. Expression levels of SPTLC1 were calculated with B2M as a reference via the ∆∆Ct method. Stable expressed reference gene for normalization was identified with RefFinder by using data from four different genes (ACTB, B2M, HPRT, and PPIA). Statistical analysis was carried out with GraphPad Prism 5 using the Mann–Whitney test.

Splicing of SPTLC1 transcripts was analyzed via amplification of the cDNA and subsequently nanopore sequencing. In brief, the cDNA from the qPCR experiments was used to amplify a part of the SPTLC1 transcript spanning the first six exons. The library was prepared with the LSK-109/EXP-NBD114 (Oxford Nanopore Technologies, Oxford, UK) library preparation kits according to the manufacturer’s protocol. The final library was loaded onto a Flongle flow cell and data was acquired for 24 h. Base calling was carried out with guppy (5.0.17). The resulting FASTQ files were aligned against die GRCh38 reference genome using minimap2 (2.24) and converted, sorted, and indexed with SAMtools (1.14). IGV (2.11.9) was used for data visualization.

## 3. Results

### 3.1. Phenotyping of a SPTLC1-Related ALS Case with Juvenile Onset

The male patient is the second child of healthy, non-consanguineous parents of German origin. His family history is unremarkable. The pregnancy was uneventful with normal fetal movements and spontaneous delivery at term with normal birth measurements. The patient was breastfed for 6 months and thrived well. At the age of 12 months, he was able to crawl coordinated, could sit without support, and tried to pull himself up. At the age of 18 months, he was able to walk without support. At the age of 3 years, it was recognized that he was not able to run. Due to additional problems affecting the fine motor abilities, school enrollment was delayed for one year. Speech and cognitive development remained normal. Chewing and swallowing were also normal.

Due to a progressive proximal weakness accompanied by difficulties in rising from the floor, a diagnostic workup was performed at the age of 7 years, including magnetic resonance imaging (MRI) of the brain and of the spinal cord indicating normal results (Figure 1A,B). Edematous alterations were identified in musculus vastus medius, lateralis, and intermedius (black arrows in Figure 1C) but no fatty degenerations were detected. An MRI of the lower legs revealed no pathological alterations (Figure 1D).

Echocardiography, standard laboratory evaluations, and a basic metabolic workup including creatine kinase indicated no abnormalities. Additionally, a muscle biopsy was performed at the age of 6 years (see below), highlighting an acute and chronic neurogenic atrophy. Based on these muscle biopsy findings, nerve conduction velocities (NCV) were investigated. However, neurography revealed normal results except for reduced right sural nerve sensory conduction velocity (23.3 m/s). The patient developed a restrictive ventilatory distress with forced vital capacity (FVC) of 53%.

Our clinical examination at the age of 7 years revealed a generalized dystrophy with a body weight of 18.4 kg (5 kg < 3rd percentile) and height of 130 cm (3rd percentile). A generalized muscular atrophy and scapulae alatae, scoliosis, and contractures of both knees and ankles were diagnosed. Patellar tendon reflexes were present. No tongue fasciculation, ataxia, or tremor were noticed. At this timepoint, he was able to walk 50 m. Due to the progressive disease course, he lost this ability to ambulate independently at the age of 12 years, and subsequently needed a spine straightening surgery at the age of 13 years. Because of the progressive bulbar dysfunction, nutrition via percutaneous endoscopic gastrostomy (PEG) tube was needed at the same time. Since the age of 13 years, he has been using a non-invasive nocturnal ventilatory support and since the age of 15 years, he also requires the use of non-invasive ventilation during daytime via mouthpiece. At this time, the patient became part of the MYO-SEQ project [14]. His last appointment at the age of 21 years revealed tetraplegia, loss of head control, and inability to lift arms or legs against gravity—he was solely able to use his fingers to drive his wheelchair via joystick.

### 3.2. Proteomic Findings in Lymphoblastoid Cells Derived from the p.(A20S)-SPTLC1-Patient

After confirming expression of SPTLC1 in blood and muscle by in silico studies (Appendix A), proteomic profiling was carried out on immortalized lymphoblastoid cells. This approach has been used in one of our previous studies, highlighting that identified protein changes in immortalized lymphoblastoid cells may provide novel insights into pathophysiological processes [15] and the previous examination of lymphoblasts derived from SPTLC1-related HSAN cases [2]. Proteomics-based quantification of 4486 proteins revealed a statistically significant increase of 13 proteins with protein tyrosine kinase 2 (PTK2; also known as focal adhesion kinase FAK1) as being the protein with, out of the decreased proteins, 11 were already linked to the manifestation of neurological diseases based on pathogenic variants in the corresponding genes (Figure 2 and Table 1).

### 3.3. Microscopic Findings on a Quadriceps Biopsy Derived from the p.(A20S)-SPTLC1-Patient

Nicotinamide adenine dehydrogenase trichrome (NADH-TR) staining revealed mild subsarcolemmal accumulation of substrate in a proportion of muscle fibers (Figure 3). The same finding was obtained by succinate dehydrogenase (SDH) stain and combined cytochrome oxidase and succinate dehydrogenase (COX-SDH) stains (Figure 3). Adenosine triphosphatase (ATPase) staining (pH 4.3—not stained = type II/dark = type I, pH 4.6—sub-typing type IIa = pale and IIb = intermediate, dark = type I, pH 9.4—pale type I/dark type II) indicated fiber-type grouping in addition to small fibers of either fiber type (Figure 3), as known pathological hallmarks of denervated muscle. Immunohistochemical studies revealed irregular Troponin-T distribution (Figure 3). Myosin heavy chain fast (MHCf) (demonstrating dark type II fibers, and conversely Myosin heavy chain slow (MHCs) demonstrating dark type I fibers illustrate both the grouping and small, often scattered or also grouped, atrophic fibers of either fiber type. Immature MHC developmental or MHC neonatal-positive fibers were apparent only scarcely.

The immunohistochemical analysis of the skeletal muscle indicated single CD3+ T cells in the endomysium and the perimysium focally (Figure 4) Few CD68+ macrophages were identified within and adjacent to an intramuscular nerve fascicle (Figure 4). Except the physiological expression within the perineurium of the nerve, no C5b-9 deposition was present (Figure 4). However, adjacent to this nerve, some single perifascicular muscle fibers indicated sarcolemmal expression of MHC-cl. I (Figure 4) which was also present on some macrophages. Moreover, macrophages expressed MHC-cl. II and some enlarged capillaries were also MHC cl. II-positive (Figure 4). Numerous small, CD56+-regenerating muscle fibers were identified (Figure 4). No cell infiltration or immune reactivity to any of the investigated molecules was observed in other areas of the biopsy. Staining of juvenile non-disease controls are added (Appendix A).

The immunofluorescence-based analysis of phospho-FAK1 on the quadricep muscle was carried out to confirm our proteomic findings, disclosing this protein as being the most increased one in patient-derived lymphoblastoid cells. In skeletal muscle, we detected expression in NDC on muscle fibers, physiological for focal adhesion. In the biopsy derived from the patient, phosphor-FAK1 was increased in perimysial areas along with increased abundance in immune cells (Figure 5).

#### Coherent Anti-Stokes Raman Scattering (CARS) Is a Nonlinear Variant of the Raman Effect

CARS allows investigations based on molecule-specific vibrations without using any dyes or labels. We used CARS to non-destructively analyze the biochemical composition of a quadricep biopsy derived from our SPTLC1-patient. Muscle fiber calibers were measured manually from CARS images utilizing the Leica LasX software. A muscle fiber caliber of 49.78 µm ± 13.17 µm on average was determined for the SPTLC1 patient. The five controls had an average fiber caliber of 43.92 µm ± 23.29 µm. Given that proteomic findings in immortalized lymphoblastoid cells indicated perturbed lipid homeostasis and increased lipid droplet accumulation was already associated with a HSAN1 [16], for the spectral analysis of biopsy sections, we focused on both wavenumbers characteristic for lipids (~2847 cm^−1^ and ~2989 cm^−1^ [17]) and proteins (2932 cm^−1^ [18]). Based on the CARS images at wave numbers 2842 cm^−1^, 2886 cm^−1^, and 2932 cm^−1^, no significant disease features, such as protein and lipid changes, could be detected (Appendix A).

### 3.4. Transcript Findings in Quadriceps Muscle Derived from the p.(A20S)-SPTLC1-Patient

To follow up on inflammatory processes in muscle suggested by our immunohistochemistry findings, we next investigated expression of transcripts that encode for markers expressed by antigen presenting cells, or markers known to either activate macrophages or be expressed by these cells. We identified increased expression of MRC1, TGFB, STAT1, STAT2, and IL1B in the patient muscle compared to NDCs by 3 to 29-fold (Table 2). A considerable increase of HLA-DRA (254-fold) and HLA-DPB (183-fold) was detected, while expression of HLA-DOB or HLA-DRB could not be detected (n.d.) at all (Table 2).

To further study SPTLC1 expression, RNA extracted from immortalized lymphocytes, derived from the patient and healthy controls, was analyzed. Our studies revealed similar expression of SPTLC1 transcripts in the patient compared to controls (Figure 6A) but decreased level of expression of the wildtype transcript containing exon 2 (Figure 6B). In line with this, nanopore sequencing of the SPTLC1 cDNA demonstrated a reduced coverage for the whole exon 2 (Figure 6C) and an additional drop in coverage for the first 28 bases of exon 2 (Figure 6D), indicating the expression of three distinct transcripts in the patient cells (Figure 6E).

## 4. Discussion

### 4.1. Clinical and Microscopic Findings of the p.(A20S)-SPTLC1-Patient

Our patient presents with a rapidly progressive motor decline which started with proximal weakness. This clinical aspect is in contrast with the phenotypical presentation of a “typical juvenile ALS disease course,” usually predominated by distal muscular atrophy, increased deep tendon reflexes, spasticity, and fasciculations [7]. Although muscle biopsy findings strongly indicated a primary neurogenic defect, NCV studies revealed almost normal findings. Given that neurogenic changes in muscle biopsies could also be found in several limb-girdle-muscular-dystrophies (LGMDs) [19], in combination with the proximal weakness in our case, a primary myopathic disease was assumed. He developed severe dysphagia and respiratory insufficiency later before he lost his ability to walk in his early teens due to the active progression. This rapid decline of motor function within the first 15 years of life is mostly seen in severe muscular dystrophies such as Duchenne muscular dystrophy, but in contrast to them, in some patients with Ullrich congenital muscular dystrophy (UCMD), non-invasive nocturnal ventilatory support may become necessary before the ability to walk is lost [20]. Increased deep tendon reflexes, spasticity, fasciculations, and the distal atrophy as typical features for juvenile ALS were not seen in our patient. The initial proximal muscular involvement and the rapid motor decline were misleading toward a severe LGMD phenotype and should thus be considered in the clinical diagnosis of juvenile ALS based on pathogenic dominant SPTLC1 variants.

### 4.2. Molecular Signature of p.(A20S)-SPTLC1-Mutant Lymphoblastoid Cells

Proteomic profiling is a robust approach to obtain unbiased insights into the pathophysiology of different diseases, including neuromuscular disorders, by utilizing different tissues or in vitro models [21,22] such as immortalized lymphoblastoid cells [15,23]. Here, we investigated the proteomic signature of immortalized lymphoblastoid cells derived from an ALS patient with juvenile-onset based on the dominant p.(A20S) variant in SPTLC1 resulting in a splicing defect, predominantly causing exon 2 skipping. Among the dysregulated proteins, SPTLC1 itself was identified with a significant decrease to 0.48-fold expression. This is the first study indicating that the presence of a dominant SPTLC1-variant leading to juvenile ALS is associated with a decrease of SPTLC1 protein level. Further transcript studies revealed no alteration of *SPTLC1* expression (Figure 6A). The expression level of the wildtype (wt) allele in the patient cells was approximately 0.5-fold compared to the healthy controls (Figure 6), indicating no compensation on RNA level for the mutated allele. Further single-molecule nanopore sequencing of the cDNA revealed, beside the skipping of exon 2 (Figure 6C), an activation of a cryptic splice site inside of exon 2 (Figure 6D,E) due to the pathogenic variant. The resulting transcript is missing the first 28 bases of exon 2 (c.58_85del), which leads to a frameshift and immediate stop (p.A20Gfs*3). However, this transcript accounts only for 5% of the total transcript amount and is very likely degraded via nonsense-mediated decay (NMD). Nevertheless, the nanopore data indicated a comparable expression of both alleles, confirming data obtained by qPCR analysis. Hence, the reduction of protein level observed by the results of our proteomic studies might originate from the decreased stability of the aberrant protein. However, functional studies are needed to address this hypothesis.

In 2013, Auer-Grumbach and co-workers reported on a French girl presenting with a syndromic phenotype defined by severe growth restriction, cognitive impairment, amyotrophy, hyperreflexia, vocal cord paralysis, and respiratory failure [24] similar to the patients described by Johnson and colleagues [7]. Although our patient did not present with cognitive impairment and had a normal brain MRI, proteomic signature of his lymphoblastoid cells unravelled a decrease of a variety of proteins associated with cognitive impairment (Table 1). Thus, one might speculate that the threshold of (i) the decrease of these proteins or (ii) the increase of compensatory proteins determines manifestation of cognitive symptoms. With respect to potential contributing proteins with decreased abundance, CTTN (decreased to 0.51-fold expression) plays a role in the regulation of neuron morphology, axon growth, and formation of neuronal growth cones [25], while DBN1 (decrease to 0.42-fold expression) is involved in memory-related synaptic plasticity in the hippocampus [26]. In addition, IQSEC1 (decrease to 0.23-fold expression) mediates the internalization of synaptic AMPAR receptors [27]. Ca^2+^ dysfunction, a common noxious mechanism in familiar and sporadic ALS, impacts on α-amino-3-hydroxy-5-methyl-4-isoxazolepropionic acid receptors (AMPA), receptor (AMPAR)-mediated excitotoxicity [28], and dysregulation of AMPAR subunit expression was already demonstrated in sporadic ALS post-mortem brains [29]. One candidate protein with potential compensatory/rescue function is PTK2, demonstrating a 13.10-fold upregulation in lymphoblastoid cells of our patient. PTK2/FAK1 acts as a non-receptor protein-tyrosine kinase and regulates axon growth and neuronal cell migration, axon branching, synapse formation, and is required for normal development of the nervous system [30]. PTK2 was linked to the protection of neurons from apoptosis [31]. Dihydropteridine reductase (QDPR; 2.69-fold increased) produces tetrahydrobiopterin (BH-4), an essential cofactor for phenylalanine, tyrosine, and tryptophan hydroxylases. Recessive QDPR variants are leading to the manifestation of severe neurologic symptoms (MIM#261630) attributable to depletion of the neurotransmitter dopamine and serotonin, whose syntheses are controlled by tryptophan and tyrosine hydroxylases that use BH-4 as cofactor. Thus, one might speculate that QDPR increase might correlate with compensatory neurotransmitter synthesis as the decrease of COMT (0.46-fold decreased), a key player of catecholamine neurotransmitters inactivation, also potentially does. PDZ and LIM domain protein 5 (PDLIM5) contributes to the regulation of dendritic spine morphogenesis in neurons (https://www.uniprot.org/uniprot/, accessed on 23 July 2018) and demonstrates a 1.97-fold increase in cells derived from our patient. Moreover, Bcl2-associated athanogene 2 (BAG2; 2.02-fold increased) acts as a regulator of p38-dependent tau kinase activity and phospho-tau degradation; tau-proteins were discussed as contributors for the manifestation of symptoms present in the clinical spectrum of SPTLC1-pathology [7,32] and are known to trigger mitophagy to protect neurons against oxidative stress [33]. Mitochondrial vulnerability was indicated to be associated with the expression of SPTLC1-variant proteins under standard culture conditions in HEK293 cells [7]. A dysregulation of mitochondrial proteins in lymphoblastoid cells derived from our patient confirms a mitochondrial vulnerability (Appendix A). Here, aldehyde dehydrogenase X (ALDH1B1; 0.46-fold decreased) represents an interesting marker, as it is involved in lipid peroxidation and the metabolism of neurotransmitters. Previous studies on lymphoblastoid cells from HSAN1-patients already displayed mitochondrial abnormalities accompanied by dysregulation of mitochondrial proteins [2,34]. However, comparative biochemical studies on lymphoblastoid cells derived from SPTLC1-related juvenile ALS patients with and without cognitive impairment are needed to prove the assumption of the modifying impact of threshold-expression of the above-discussed proteins in the clinical manifestation of diverse symptoms of SPTLC1-pathology.

Taking the subcellular localization of SPTLC1 within the ER into consideration, previous studies on lymphocytes derived from HSAN1-patients reported notable changes in ER homeostasis associated with increase of ER-stress marker proteins, such as binding immunoglobulin protein (BiP) [2]. Our proteomic data did not hint toward unfolded protein response (UPR) activation in terms of an identified elevation of related proteins. In contrast, proteomic signature of our patient-derived cells indicated a decrease of ER-resident proteins such as protein O-glucosyltransferase 3 (POGLUT3; 0.46-fold decreased), a protein glucosyltransferase catalysing the transfer of glucose from uridine diphosphate (UDP)-glucose to a serine residue [35], and inositol 1,4,5-trisphosphate receptor type 2 (ITPR2; 0.41-fold decreased), a receptor for inositol 1,4,5-trisphosphate, a second messenger that mediates the release of intracellular Ca2+, in turn providing a direct biochemical link to perturbed Ca2+ homeostasis as a well-known pathomechanism in the etiology of ALS [28]. Moreover, phosphatidylinositol glycan anchor biosynthesis class K (PIGK) along with phosphatidylinositol glycan anchor biosynthesis class S (PIGS) are decreased ER-resident proteins. Both are involved in glycosylphosphatidylinositol-anchor biosynthesis, which is part of glycolipid biosynthesis. A role of glycosphingolipids in neuronal polarity was already demonstrated in a C. elegans model of HSAN1 [36]. Further hints of perturbed lipid homeostasis are given by decrease of diacylglycerol kinase epsilon (DGKE; 0.42-fold decrease) that regulates levels of bioactive lipids [37,38,39,40,41], and of N-myc downstream regulated family member 1 (NDRG1; 0.31-fold decreased), which functions in lipid metabolism [42]. Recessive NDRG1-variants were linked to manifestation of a demyelinating neuropathy [43].

Interestingly, GO-term based analysis of our proteomic findings obtained in immortalized lymphoblastoid cells displayed retina homeostasis as an affected (up-regulated) biological process and, in 2019, retinal disease was reported in patients carrying SPTLC1 variants [44]. Further up-regulated processes are indicative for activation of the immune response, and immunoglobulin Ig kappa up-regulation was already identified by proteomic investigations on lymphoblastoid cells of HSAN1-patients [34], and could moreover be identified in cells of our patient (3.90-fold increase).

### 4.3. Findings in the Quadriceps Biopsy of the p.(A20S)-SPTLC1-Patient

Microscopic studies revealed fiber-type grouping, a very well-known pathomorphological hallmark of denervated skeletal muscle, accompanied by subsarcolemmal increase of mitochondria in a proportion of muscle fibers. This latter finding is rather nonspecific and has already been described in the context of muscle cell denervation [45]. MyHC-neo and -developmental are predominantly expressed in developing muscle fibers [46] and are not increased in the muscle biopsy of our patient, suggesting absence of immature myofibers. Our immunohistochemical studies revealed irregular Troponin-T expression and Troponins have been postulated to serve as a good protein marker for muscle pathology based on loss of motor neuron function [47]. Although excess of sphingolipid synthesis is likely the major pathomechanism caused by the gain of the function variant identified in our patient [6], our CARS microscopic investigations did not indicate the presence of lipid aggregates (and Oil-red-O did not indicate increased lipid droplets). This might be based on a functional irrelevance of SPTLC1 in muscle (SPTLC1 expression is low in muscle compared to nervous tissue, blood, and cultured fibroblasts) and/or the activation of compensatory mechanisms. However, more functional studies on muscle cells are needed to prove this assumption and to delineate a potential functional role of SPTLC1 in muscle cells. In addition, more detailed studies on nervous tissue are needed to investigate if the gain of SPTLC1-function leads to lipid aggregation.

Although immune system components may indeed play a role in ALS pathogenesis, detailed studies on their impact in early disease pathology are limited, and more focused studies examining the role of the immune system together with a characterized pathogenesis to determine when, where, and if immune and inflammatory processes are taking place are still needed [48]. Here, combined immunohistochemistry and transcript studies revealed increased antigen presentation via the major histocompatibility complex within a small area in which immune cells (macrophages) are also detectable, indicating an active (while still limited) immune response. HLA class II antigens play a central role in the immune system by presenting peptides derived from extracellular proteins and are expressed in antigen presenting cells, like macrophages. Although an elevated expression accords with our histological findings, the very high transcript level of the HLA-DRA and HLA-DRB, respectively, remain surprising, given that the overall immunoreactivity for these proteins was of minor increase in the patient-derived muscle biopsy. This finding indicates that increased expression of the *HLA-DRA* and *HLA-DRB* genes does not necessarily correlate with the same extension of an immune response on the protein level, and that combined transcript and protein studies are needed to draw a conclusion with respect to the intensity of respective protein contributions on muscle inflammation.

Our confirmational studies focussing on phospho-FAK1 abundance and distribution in quadriceps muscle of the patient revealed increased expression within the perimysium. The focal adhesion kinase (FAK1) is known to act as a downstream non-receptor tyrosine kinase that translates cytoskeletal stress and strain signals transmitted across the cytoplasmic membrane to activate multiple anti-apoptotic and cell growth pathways. Notably, changes in FAK expression and phosphorylation correlate to specific developmental states in myoblast differentiation, muscle fiber formation and muscle size in response to loading and unloading [49]. However, the increase within indicates a role in fibrosis, and a model wherein hypoxia (as a secondary insult) caused exacerbated pulmonary fibrosis elevated expression of phosphorylated FAK1 was already described [50].

## 5. Conclusions

First manifestation of proximal muscle weakness should be considered in the clinical diagnosis of SPTLC1-related juvenile ALSP.(A20S)-SPTLC1-related ALS leads to muscle denervation accompanied by fiber-type grouping, subsarcolemmal accumulation of mitochondria in type I fibers, and mild inflammationAggregates of proteins or lipids were not identified in patient-derived quadriceps muscle by CARS microscopyProteomic profiling on p.(A20S)-SPTLC1-mutant lymphoblastoid cells unraveled a variety of proteins with potential impact on clinical manifestation of symptoms of SPTLC1-related phenotypical spectrumFurther functional in vitro studies are needed to systematically address the impact of these proteins in the pathogenesis of increased SPTLC1 functionThe molecular genetic SPTLC1 variant identified in our patient is associated with altered splicing and decrease of protein levelIncreased expression of phosphor-FAK1 in patient-derived quadriceps muscle might contribute to fibrotic degeneration

## Figures and Tables

**Figure 1 genes-13-00893-f001:**
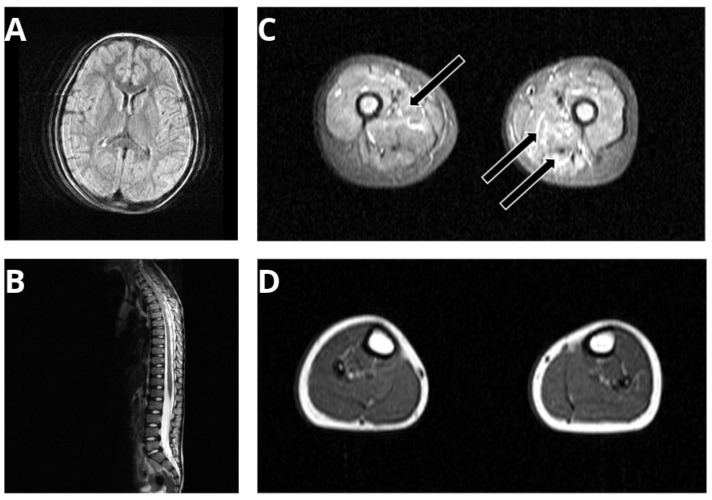
Clinical findings in the p.(A20S)-SPTLC1-patient: MRI of brain (**A**) and spinal cord (**B**) revealed normal findings. Edematous alterations in musculus vastus medius, lateralis, and intermedius (black arrows in **C**). No fatty degenerations were detected, and MRI of lower legs revealed no pathological alterations (**D**). The limited quality of the MRI images is based on motion artifacts of the child during the investigation.

**Figure 2 genes-13-00893-f002:**
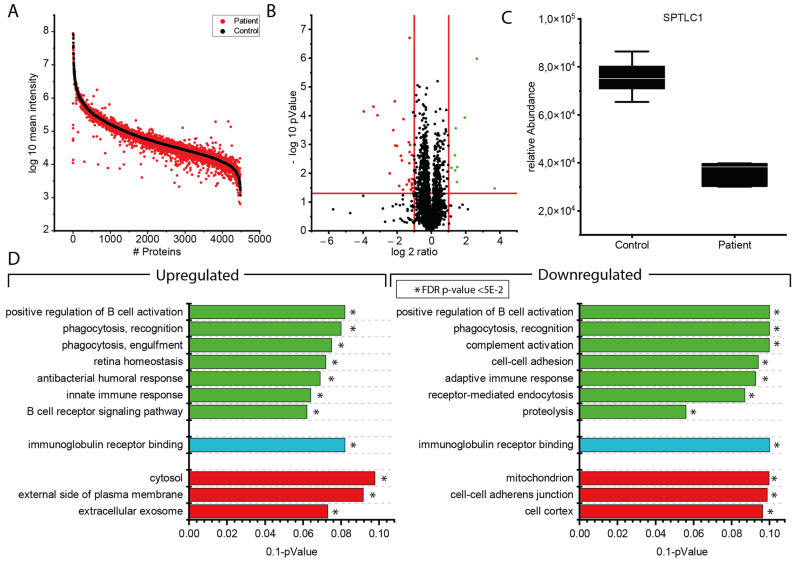
Proteomic findings in immortalized lymphoblastoid cells derived from the p.(A20S)-SPTLC1-patient: (**A**) Protein enrichment plot along the entire abundance range. Proteins highlighted in black represent the control results, sorted by abundance. Proteins highlighted in red represent the abundance of the patient proteins, left in the same order as the control proteins to highlight differences in their expression. (**B**) Volcano-plot of proteomic findings. Red dots represent proteins demonstrating a significant decrease whereas green dots represent proteins with a significant increase in abundance. Red lines indicate the significance thresholds of *p*-value 0.05 (*y*-axis) and 2-fold increase or 0.5-fold decrease in abundance (*x*-axis). (**C**) Boxplot depicting the relative abundance of the SPTLC1 protein comparing control- and patient-derived samples. (**D**) GO-term based pathway analysis for up- and downregulated biological processes (green), molecular functions (blue), and cellular compartments (red) which are significantly regulated by comparing control and patient samples. Terms are sorted from top to bottom with decreasing level of significance. To achieve a better representation, the *x*-axis is given as 0.1 *p*-value (* = statistically significant).

**Figure 3 genes-13-00893-f003:**
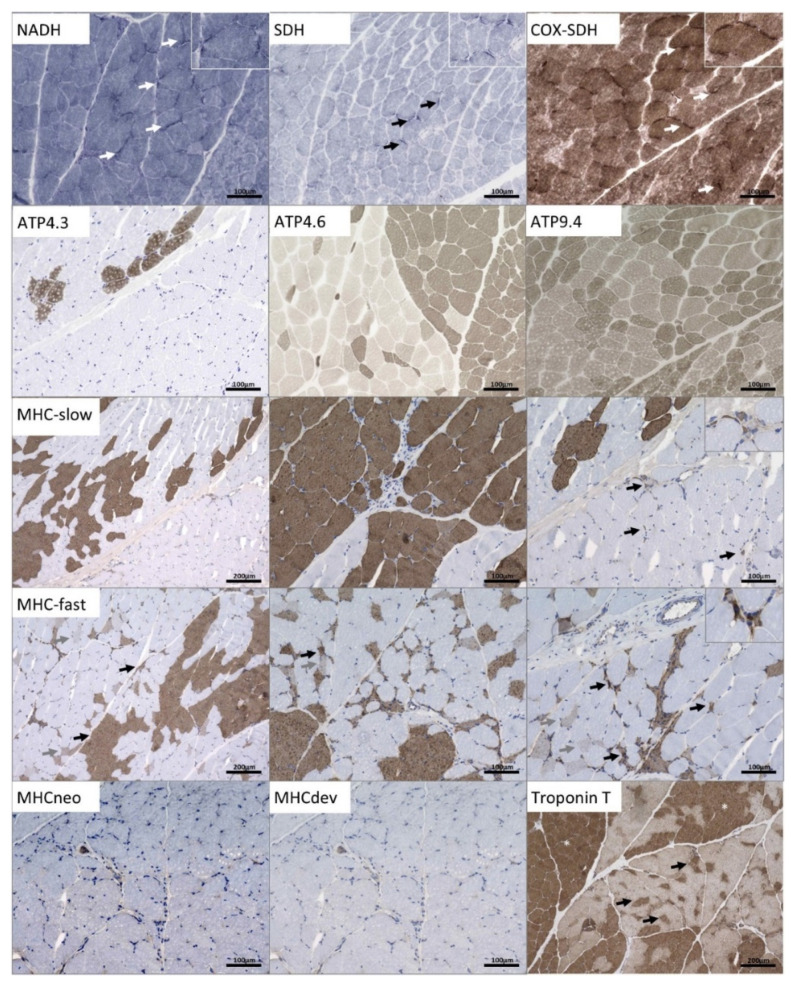
Light microscopic (histology and immunohistology) findings in a quadricep muscle biopsy derived from the p.(A20S)-SPTLC1-patient: Enzyme histochemical NADH-TR, SDH, and COX-SDH preparations indicating mild subsarcolemmal accumulation of substrate in single fibers (arrows). Grouping and atrophy of fibers of either fiber types are highlighted with ATPase preparations incubated at different pH levels, like the immunohistochemical MHC fast and –slow stains (arrows indicating atrophic type I and II fibers. Only single immature fibers expressing MHC-developmental or –neonatal. Troponin T indicates some mildly irregular staining on atrophic fibers (arrows).

**Figure 4 genes-13-00893-f004:**
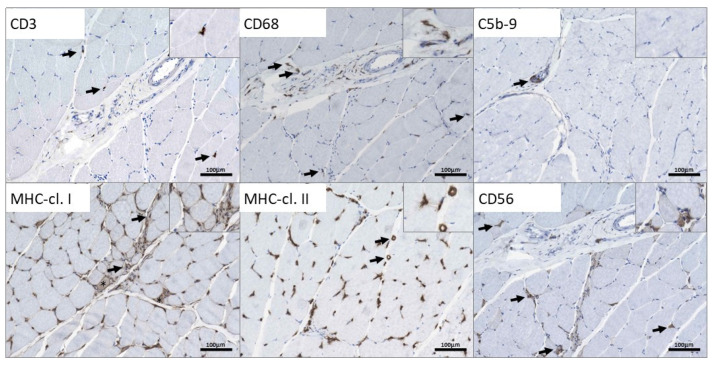
Immunohistological studies focusing on inflammatory proteins in a quadricep muscle biopsy derived from the p.(A20S)-SPTLC1-patient: CD3 staining highlights single T cells in the endomysium and the perimysium (arrows). CD68+ macrophages are diffusely scattered in the endomysium and accumulate around and within a small perimysial nerve fascicle (arrows). Physiological complement accumulation in the perineurium of this nerve (arrow), but not on any sarcolemmal or sarcoplasmic structure of muscle fibers. Single myofibers in perifascicular regions indicate upregulation of MHC cl. I sarcolemmaly (asterix), and some macrophages in the perimysium are also MHC cl. I-positive (arrows). MHC cl. II similarly highlights some macrophages and the partially enlarged capillary vessels (arrows). Small regenerating fibers also express CD56 (arrows).

**Figure 5 genes-13-00893-f005:**
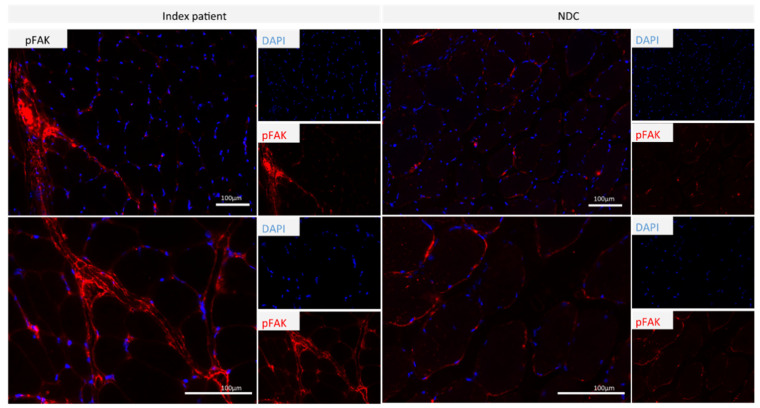
Immunofluorescence studies focusing on phosphor-FAK1 (pFAK) in quadriceps muscle biopsies derived from the p.(A20S)-SPTLC1-patient and a control: phospho-FAK1 is expressed in the sub-sarcolemmal region of muscle fibers derived from the healthy control, whereas in the biopsy derived from the patient, phospho-FAK1 is upregulated in perimysial areas and in immune cells.

**Figure 6 genes-13-00893-f006:**
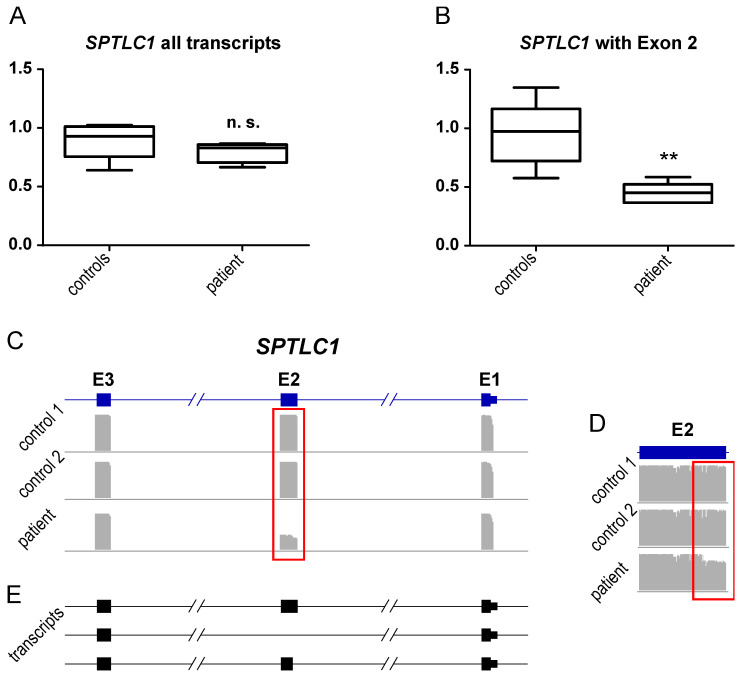
SPTLC1 expression in lymphocytes: RNA was extracted from immortalized lymphocytes derived from the patient and healthy controls. cDNA was generated with random hexamer primers to perform expression analysis of SPTLC1. mRNA expression was calculated via ∆∆Ct method using B2M for normalization. The SPTLC1 expression was normalized to the healthy controls. Boxes indicate the lower and upper interquartiles, horizontal lines indicate the median, and whiskers indicate the minimum-to-maximum values. ** *p* < 0.01, n. s. not significant, by Mann–Whitney test. (**A**) SPTLC1 expression of all transcripts. (**B**) SPTLC1 expression of transcripts containing exon 2. cDNA from healthy controls and the patient were amplified and used for ligation-based nanopore library preparation and sequencing. IGV coverage plots of SPTLC1 exons 1–3 (**C**) or exon 2 (**D**), respectively. (**E**) Compositions of the different SPTLC1 transcript detected by nanopore sequencing in the patient.

**Table 1 genes-13-00893-t001:** Proteins decreased in the proteomic signature of p.(A20S)-SPTLC1-mutant lymophoblastoid cells associated with neurological diseases.

Protein Accession	Gene Name	Protein Name	Unique Peptides	Fold of Decrease	*p*-Value	Neurological Disease	OMIM
Q9HA77	CARS2	Probable cysteine-tRNA ligase, mitochondrial	9	0.54	0.000	Combined oxidative phosphorylation deficiency 27	COXPD27; MIM:616672
Q9NVH6	TMLHE	Trimethyllysine dioxygenase, mitochondrial	2	0.54	0.004	Autism, X-linked 6	AUTSX6; MIM:300872
Q9ULJ6	ZMIZ1	Zinc finger MIZ domain-containing protein 1	2	0.51	0.019	Neurodevelopmental disorder with dysmorphic facies and distal skeletal anomalies (NEDDFSA; MIM:618659)	NEDDFSA; MIM:618659
Q92643	PIGK	GPI-anchor transamidase	2	0.50	0.016	Neurodevelopmental disorder with hypotonia and cerebellar atrophy, with or without seizures	NEDHCAS; MIM:618879
Q96S52	PIGS	GPI transamidase component PIG-S	5	0.49	0.002	Glycosylphosphatidylinositol biosynthesis defect 18	GPIBD18; MIM:618143
P04792	HSPB1	Heat shock protein β-1	13	0.49	0.050	Charcot-Marie-Tooth disease 2F & Neuronopathy, distal hereditary motor, 2B	CMT2F; MIM:606595 & HMN2B; MIM:608634
P21964	COMT	Catechol O-methyltransferase	3	0.46	0.032	Schizophrenia	SCZD; MIM:181500
Q16643	DBN1	Drebrin	9	0.42	0.000	Alzheimer disease	AD; MIM:104300
Q92597	NDRG1	Protein NDRG1	2	0.31	0.004	Charcot-Marie-Tooth disease 4D	CMT4D; MIM:601455
Q6DN90	IQSEC1	IQ motif and SEC7 domain-containing protein 1	3	0.23	0.000	Intellectual developmental disorder with short stature and behavioral abnormalities	IDDSSBA; MIM:618687
Q7L3T8	PARS2	Probable proline-tRNA ligase, mitochondrial	2	0.19	0.010	Developmental and epileptic encephalopathy 75	DEE75; MIM:618437

**Table 2 genes-13-00893-t002:** Transcript expression in the p.(A20S)-SPTLC1-patient derived quadricep muscle.

HLA-DRA	HLA-DPB1	HLA-DOB	HLA-DRB	TGFB	STAT1	STAT2	IL1B	MRC1 (CD206)
254×	183×	n.d.	n.d.	28×	7×	29×	27×	3×

Values are depicted as fold-change in the patient sample versus non-disease control level. n.d. = not detectable.

## Data Availability

The data that support the findings of this study are available from the corresponding author upon reasonable request.

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
