# Peer review of "New Insights into the Neuromyogenic Spectrum of a Gain of Function Mutation in SPTLC1"

_genes, 2022, doi:10.3390/genes13050893_

Round 1

Reviewer 1 Report

The SPTLC1 gene is associated with juvenile ALS, a fatal neurological disorder whose pathology is still to be determined. Because of the lack of molecular and biochemical understanding of its pathophysiology, no cure is available. In this manuscript, Kölbel H. et al, evaluate the impact of p.(A20S)-SPTLC1 variant on the proteomic profile of lymphoblastoid cells and demonstrate the histochemical changes on mitochondrial markers and inflammatory proteins in a quadriceps biopsy derived from the p.(A20S)-SPTLC1-patient. We believe that the manuscript is well written and the conclusions are in general supported by these experiments, however, we have some comments.

Concerns and suggestions:

  1. In the manuscript, we are concerned about the fact that the authors only include one SPTLC1-patient suffering from a juvenile-onset of ALS, and no in vitro studies were done. The authors should use in vitro system to a) support the impact of this point mutation p.(A20S)-SPTLC1 on its mRNA and protein expression b) explore the association of this mutation and the sub-cellular localization of SPTLC1 protein c) assess the function of SPTLC1 and preform Western blotting validation of its downstream genes resulted from the proteomic signature of p.(A20S)-SPTLC1.  
  2. Suppl. Figure 1 is not provided.  
  3. Images in Figure 2 are low quality (very blurry). 
  4. Show the data in Table 2 as boxplots, similar to Figure 5A and 5B.  
  5. The authors used immunohistochemistry to demonstrate the changes within the quadriceps biopsy derived from the p.(A20S)-SPTLC1-patient without showing tissue from healthy control (Fig.3 and 4). 
  6. A typo in line 386 (without using without any dyes or labels), without is repeated twice. 

Author Response

Comments and Suggestions for Authors Reviewer1:

The SPTLC1 gene is associated with juvenile ALS, a fatal neurological disorder whose pathology is still to be determined. Because of the lack of molecular and biochemical understanding of its pathophysiology, no cure is available. In this manuscript, Kölbel H. et al, evaluate the impact of p.(A20S)-SPTLC1 variant on the proteomic profile of lymphoblastoid cells and demonstrate the histochemical changes on mitochondrial markers and inflammatory proteins in a quadriceps biopsy derived from the p.(A20S)-SPTLC1-patient. We believe that the manuscript is well written and the conclusions are in general supported by these experiments, however, we have some comments.

Concerns and suggestions:

Point of concern 1: In the manuscript, we are concerned about the fact that the authors only include one SPTLC1-patient suffering from a juvenile-onset of ALS, and no in vitro studies were done. The authors should use in vitro system to a) support the impact of this point mutation p.(A20S)-SPTLC1 on its mRNA and protein expression b) explore the association of this mutation and the sub-cellular localization of SPTLC1 protein c) assess the function of SPTLC1 and preform Western blotting validation of its downstream genes resulted from the proteomic signature of p.(A20S)-SPTLC1.  

Reply:  We agree with this important point of concern: optimally in vitro studies are needed to support our hypothesis. However, a considerable amount of these studies were done prior to the study presented here (which also has a profound focus muscle pathology) and were published by Mohassel P. (Childhood amyotrophic lateral sclerosis caused by excess sphingolipid synthesis) in Nat Med. 2021 Jul;27(7):1197-1204. doi: 10.1038/s41591-021-01346-1. Epub 2021 May 31. PMID: 34059824. To address the point of concern, we added this information within the introduction section, page 2, lines 64-67.

Point of concern 2: Suppl. Figure 1 is not provided.  

Reply:  We thank the reviewer for this remark. We add the supplemental material at the end of the manuscript pages 23-27.

Point of concern 3: Images in Figure 2 are low quality (very blurry). 

Reply:  We thank the reviewer for pointing out the low quality of this figure. Unfortunately, the quality was influenced by the movement artefact of the patient/child (7 years of age). We add a statement in the figure’s legend (page 6, lines 257-258).

Point of concern 4: Show the data in Table 2 as boxplots, similar to Figure 5A and 5B. 

Reply: We thank the reviewer for this remark. Unfortunately, the transcript expression in the p.(A20S)-SPTLC1-patient-derived quadriceps muscle was confirmed in one measurement, so a boxplot cannot be composed.

Point of concern 5: The authors used immunohistochemistry to demonstrate the changes within the quadriceps biopsy derived from the p.(A20S)-SPTLC1-patient without showing tissue from healthy control (Fig.3 and 4). 

Reply: We thank the reviewer for this remark and add health control muscle in figures 3 and 4 as the Supplemental figure S2 page 24.

Point of concern 6: A typo in line 386 (without using without any dyes or labels), without is repeated twice. 

Reply:  We thank the reviewer for pointing out the typo. We deleted the duplication.  

Reviewer 2 Report

SPTLC1 mutations have been recently described through large genetic screens as associated to juvenile forms of the progressive neurodegenerative disease amyotrophic lateral sclerosis (ALS). SPTLC1 is a component of the serine palmitoyltransferase, a key enzyme in the sphingolipid metabolism. In this manuscript, Kolbel et al. describe clinical history of a juvenile ALS patient carrying a A20S mutation in SPTLC1. The authors performed proteomic analysis from lymphoblastoid cells derived from the affected patients, as well as immunohistochemical examination of a muscle biopsy, with the aim to further our understanding of the biochemical and cellular mechanisms linking SPTLC1 and ALS.

While the topic is interesting and timely, there are several concerns about the lack of rigor for some of the experiments, the overinterpretation of the results, and the preliminary nature of most of the data that greatly diminish the enthusiasm for the study.

  • There is no mention in either the result section or the methods section, to what control cells were used for the proteomics assays, as well as for the transcriptional analysis of muscle biopsy. Without that information it is extremely hard to interpret any of the data presented. It is also unclear how many biological or technical replicates were used for any of the experiments described.
  • The clinical description of the patient is detailed but it is unclear when and how the SPTLC1 mutation was identified.
  • The proteomics data obtained from lymphoblast cells appear to not have been validated by any other method. Further, no data was presented to assess the relevance of such changes in the muscle tissue. For instance, the authors note that PTK2 was the most upregulated protein in the dataset. If would have been interesting to assess whether a similar increase was also detected in the muscle biopsy.
  • A long discussion about the interpretation of the proteomics data, including a reference to several up or downregulated proteins associated with neurological diseases or potential protective factors, is unwarranted given the preliminary nature of the data described. A similar concern applies to the data showing increase inflammation in the muscle tissue, which appears to be mild.
  • The quality of the images, particularly Figure 2, is poor and it is impossible to read any labels in the graphs. Supplementary materials were not provided to this reviewer.

Author Response

Comments and Suggestions for Authors Reviewer2

SPTLC1 mutations have been recently described through large genetic screens as associated to juvenile forms of the progressive neurodegenerative disease amyotrophic lateral sclerosis (ALS). SPTLC1 is a component of the serine palmitoyltransferase, a key enzyme in the sphingolipid metabolism. In this manuscript, Kolbel et al. describe clinical history of a juvenile ALS patient carrying a A20S mutation in SPTLC1. The authors performed proteomic analysis from lymphoblastoid cells derived from the affected patients, as well as immunohistochemical examination of a muscle biopsy, with the aim to further our understanding of the biochemical and cellular mechanisms linking SPTLC1 and ALS.

While the topic is interesting and timely, there are several concerns about the lack of rigor for some of the experiments, the overinterpretation of the results, and the preliminary nature of most of the data that greatly diminish the enthusiasm for the study.

Concerns and suggestions:

Point of concern 1: There is no mention in either the result section or the methods section, to what control cells were used for the proteomics assays, as well as for the transcriptional analysis of muscle biopsy. Without that information it is extremely hard to interpret any of the data presented. It is also unclear how many biological or technical replicates were used for any of the experiments described.

Reply: We thank the reviewer for this remark and the opportunity to improve the method section. For proteomic analysis, samples from two healthy individuals and the sample from the patient were processed. After the preparation of immortalized lymphoblastoid (see 2.1), samples were propagated (cell culture) to 2x 3 samples for controls and 1x 3 samples for the patient. In total, we processed 9 samples independently in our proteomics sample preparation and analysis pipeline (page 4, lines 143-144). For the studies on muscle, two age- and sex-matched controls were used, and representative data are now included for one of these controls.

Point of concern 2: The clinical description of the patient is detailed but it is unclear when and how the SPTLC1 mutation was identified.

Reply: We thank the reviewer for this remark. We now added the information of the inclusion of the patient in the MYO-SEQ project at the age of 15 years on page 6, lines 277-278.

Point of concern 3: The proteomics data obtained from lymphoblast cells appear to not have been validated by any other method. Further, no data was presented to assess the relevance of such changes in the muscle tissue. For instance, the authors note that PTK2 was the most upregulated protein in the dataset. If would have been interesting to assess whether a similar increase was also detected in the muscle biopsy.

Reply: We thank the reviewer for this remark and the opportunity to improve the manuscript by performing verification studies for PTK2/ FAK1 in skeletal muscle. Results are now added as Figure.5 in the result section page 12, lines 383-389. We moreover included interpretations of FKA1-related findings within the discussion section page 17, lines 611-620.

Point of concern 4: A long discussion about the interpretation of the proteomics data, including a reference to several up or downregulated proteins associated with neurological diseases or potential protective factors, is unwarranted given the preliminary nature of the data described. A similar concern applies to the data showing increase inflammation in the muscle tissue, which appears to be mild.

Reply: We thank the reviewer for this essential comment. We want to point out that the evaluation of one patient-derived muscle of this ultra-rare disease leads to many speculative results, which should be discussed in extension. Our aim is to enable other researcher to confirm these ideas by further investigations. A respective comment is now made in the conclusion section of the revised version of the manuscript. We moreover consider the description of mild inflammation as an important aspect regarding the diagnostic management of future SPTLC1 patients.

Point of concern 5: The quality of the images, particularly Figure 2, is poor and it is impossible to read any labels in the graphs.

Reply:  We thank the reviewer for pointing out the low quality of these figures. Unfortunately, the quality of figure 1 was influenced by the movement artefact of the patient/child (8 years of age). We add a statement in the figure’s legend (page 6, lines 257-258) and we enlarged Figure 2.

Point of concern 6: Supplementary materials were not provided to this reviewer.

Reply:  We thank the reviewer for this remark. We add the supplemental material at the end of the manuscript page 23-27.

Round 2

Reviewer 1 Report

Thank you for revising the paper in accordance with the comments. I have a few comments on the analysis and presentation.

  • Supplemental Table S1 needs a caption. Also, some P values in the table are equal to 0.000, please fix them to a proper value.
  • Supplemental Figure S3 needs more information in the caption. Use arrows to indicate the protein of interest wherever it is possible.
  • Nearly all figures/images are low quality (blurry) in your pdf compared to word. It is very important to resolve this issue. 

Good luck

Author Response

Comments and Suggestions for Authors Reviewer1:

Point of concern: Supplemental Table S1 needs a caption. Also, some P values in the table are equal to 0.000, please fix them to a proper value.

Reply: We thank the reviewer for this remark and add the caption: Expression of the serine palmitoyltransferase (SPT) complex at various abundances across the neuromuscular axis and within cultured fibroblasts and blood cells (page 27, lines 904-905) and we fix a proper value for all P values 0,000 as <0,0005.

Point of concern: Supplemental Figure S3 needs more information in the caption. Use arrows to indicate the protein of interest wherever it is possible.

Reply: We thank the reviewer for this remark. We performed histological and immunohistological analyses on quadriceps muscle derived from juvenile non-disease controls as recommended in the Supplemental figure S2 and add this information in the caption. We also corrected the typo from S3 to S2 (page 24, lines 882-883).

In Supplemental Figure S3 we add in the caption Coherent anti-Stokes Raman scattering (CARS) images of the p.(A20S)-SPTLC1 patient-derived quadriceps muscle.

Unfortunately, no lipid or protein accumulations could be detected at the wavenumbers 2842 cm-1 (lipid), 2886 cm-1 (lipid) and 2932 cm-1 (protein), so we cannot add arrows (page 25, lines 894-895).

Point of concern: Nearly all figures/images are low quality (blurry) in your pdf compared to word. It is very important to resolve this issue. 

Reply: The transformation from word to pdf was automatically done from the upload system from MPDI and unfortunately cannot change by the authors.

Reviewer 2 Report

The authors have done a reasonable job at addressing the previous concerns. The quality of Figure 2 and Figure 5 is still not good, but that could be an issue with the pdf conversion since the images in the docx version of the supplementary data are significantly better than the ones in the pdf. The description of the control lines used for the proteomics analysis (i.e. age and gender of donor) must be included in the results and/or methods section. The authors have only added the total number of samples processed in the revised manuscript.

Author Response

Comments and Suggestions for Authors Reviewer 2

Point of concern 1: The quality of Figure 2 and Figure 5 is still not good, but that could be an issue with the pdf conversion since the images in the docx version of the supplementary data are significantly better than the ones in the pdf.

Reply: We thank the reviewer for this remark. The transformation from word to pdf was automatically done by the upload system from MPDI and unfortunately cannot change by the authors.

Point of concern 2: The description of the control lines used for the proteomics analysis (i.e. age and gender of donor) must be included in the results and/or methods section.

The authors have only added the total number of samples processed in the revised manuscript.

Reply: We thank the reviewer for this remark and add the additional information in the method section

(page 4 lines 143-146 and lines 179-180).
